# Lipidomics in Understanding Pathophysiology and Pharmacologic Effects in Inflammatory Diseases: Considerations for Drug Development

**DOI:** 10.3390/metabo12040333

**Published:** 2022-04-07

**Authors:** Kabir Ahluwalia, Brandon Ebright, Kingsley Chow, Priyal Dave, Andrew Mead, Roy Poblete, Stan G. Louie, Isaac Asante

**Affiliations:** 1School of Pharmacy, University of Southern California, Los Angeles, CA 90089, USA; kahluwal@usc.edu (K.A.); bebright@usc.edu (B.E.); chowking@usc.edu (K.C.); priyalda@usc.edu (P.D.); amead@usc.edu (A.M.); 2Department of Neurology, Keck School of Medicine, University of Southern California, Los Angeles, CA 90033, USA; roy.poblete@med.usc.edu; 3Department of Ophthalmology, Keck School of Medicine, University of Southern California, Los Angeles, CA 90033, USA

**Keywords:** lipidomics, inflammation, eicosanoids, mass spectrometry, cardiovascular disease, neurodegeneration, autoimmune disease, respiratory disease, special pro-resolving lipid mediators, lipid mediators

## Abstract

The lipidome has a broad range of biological and signaling functions, including serving as a structural scaffold for membranes and initiating and resolving inflammation. To investigate the biological activity of phospholipids and their bioactive metabolites, precise analytical techniques are necessary to identify specific lipids and quantify their levels. Simultaneous quantification of a set of lipids can be achieved using high sensitivity mass spectrometry (MS) techniques, whose technological advancements have significantly improved over the last decade. This has unlocked the power of metabolomics/lipidomics allowing the dynamic characterization of metabolic systems. Lipidomics is a subset of metabolomics for multianalyte identification and quantification of endogenous lipids and their metabolites. Lipidomics-based technology has the potential to drive novel biomarker discovery and therapeutic development programs; however, appropriate standards have not been established for the field. Standardization would improve lipidomic analyses and accelerate the development of innovative therapies. This review aims to summarize considerations for lipidomic study designs including instrumentation, sample stabilization, data validation, and data analysis. In addition, this review highlights how lipidomics can be applied to biomarker discovery and drug mechanism dissection in various inflammatory diseases including cardiovascular disease, neurodegeneration, lung disease, and autoimmune disease.

## 1. Introduction

Phenotypic lipid profiling techniques have dramatically improved in the last decade. This technological advancement has ushered in a movement towards multianalyte identification and quantification of endogenous lipids and their metabolites which are potent regulators of inflammation and pro-resolving processes. The ability to comprehensively understand the metabolism of these important structural components and cellular signaling molecules has deepened our insights into disease pathophysiology and helped identify drug targets for potential therapeutic development. Additionally, lipidomic profiling has identified imbalances in lipid homeostasis in several diseases of global health concern including metabolic syndrome, cardiovascular disease (CVD), neurodegenerative disease, respiratory disease, and autoimmune disease [1,2]. Importantly, these changes in the lipidome appear to occur at specific stages during disease progression, where lipidomics can be applied to identify reliable biomarkers to monitor disease severity [3,4]. In this regard, the ability to realign the lipidome may also serve as a good indicator of the efficacy of therapeutic interventions.

Metabolomics is an analytical approach that utilizes mass detection systems to provide a cross-sectional snapshot of the metabolites involved in a particular biological system in order to dissect metabolic pathways as well as quantify analytes of interest. Lipidomics represents a category of metabolomics centered on the metabolomic evaluation of lipids and their metabolites. While lipids are known to be potent mediators of inflammation and other cell signaling cascades, lipidomics continues to be underutilized in guiding our understanding of disease pathophysiology and the impact of therapeutic interventions.

As clinical trials have become more reliant on biomarkers as disease modifying endpoints, lipidomic monitoring may be highly applicable in drug development. Currently, there is a lack of guidance as to how lipidomics can be deployed to understand pharmacological interventions, and how it can be used to facilitate safety and efficacy assessments in clinical trials. To address this lack of standardized workflows for lipidomics, the National Institute of Standards and Technology (NIST) conducted surveys and studies of laboratories in pursuit of creating harmonization and improved standardization of lipidomic methodologies and findings [5,6]. This review will highlight some of the strengths and limitations associated with the current approaches of lipidomics. Key considerations for lipidomics include data acquisition and analysis, as well as how lipidomics can guide the dissection of the pharmacological effects of interventions.

## 2. Techniques and Approaches to Lipidomics

The ability to simultaneously detect multiple analytes has been denoted as “Omics” approaches. This powerful technique combines layered information derived from genetics, transcriptomics, and proteomics, where metabolic data can improve our understanding of the molecular mechanisms driving disease progression. This approach can be globally assessed to identify changes in biomatrices as well as tissues, where correlations can be made between analyte concentrations and other measures of disease severity. Accurate correlations with disease severity highlight the power of this approach that can be transformed into diagnostic biomarkers [7].

Although proteins and nucleic acids have been thoroughly evaluated using proteomics and genomics, respectively, broad spectrum determination of genes and proteins may be limited due to a lack of contextualization. Metabolomics/lipidomics has the advantage of relating phenotype to alterations in an entire metabolic pathway, such as increased or decreased enzymatic activities. Metabolomic/lipidomic approaches, which can determine either the relative levels or precise concentrations of targeted analytes within a specific biomatrix, are most important where gene expression cannot determine the levels of active metabolites. Examples of this would include structural lipids (e.g., phospholipids) found in the cellular membranes which can be metabolized to form bioactive lipids capable of regulating or promoting the resolution of inflammation. The ability to quantify levels of precursor molecules and their metabolites provides a more complete picture of the relationships between molecular changes and phenotypical outcomes.

Inflammatory responses are mediated by various chemical entities including histamine, eicosanoid lipids, and cytokines [8,9]. Metabolomic and lipidomic approaches require greater implementation to capture a more robust profile of active metabolites involved in inflammatory cascade pathways. Lipidomic techniques have advanced with instrumentation innovation allowing for accurate multianalyte analyses [10]. Analyses can differ on several levels including sample biomatrices, sample preservation, sample processing, types of internal standards, instrumentation, untargeted and targeted analysis, and data analysis. The following sections will review lipidomic instrumentation, lipid stabilization, data validation considerations, and data analysis approaches.

### 2.1. Instrumentation

At the center of these metabolomics/lipidomics analyses is the use of mass spectrometry (MS) technology [11]. This powerful tool can identify and quantify a wide spectrum of analytes simultaneously. Nuclear Magnetic Resonance (NMR) spectroscopy is yet another tool that can be used for metabolomics/lipidomics analyses. While NMR is an alternative approach with better resolution to elucidate molecular structures, large sample volumes and micromolar concentrations are required for this approach [12]. In contrast, MS-based techniques can identify and quantify lipid analytes at nanomolar or even picomolar concentrations [13]. Current uses of MS analyses have employed either untargeted or targeted approaches [14,15]. Untargeted or “shotgun” approaches are usually employed to identify unknown metabolites that may participate in a biological system. A limitation of this approach is that it is incapable of precise analyte identification, and thus requires additional tests to confirm analyte structures. In contrast, targeted analyses aim to quantify known analytes with previously defined structures.

Currently there is a lack of uniformity in the types of MS technologies employed. The diversity of MS methods used includes liquid chromatography (LC)-triple quadrupole MS, LC-quadrupole time-of-flight MS (LC-QTOF/MS), LC-Orbitrap, nano-electrospray ionization MS (nano-ESI–MS), gas chromatography MS (GC–MS), acoustic ejection MS, and matrix-assisted laser desorption ionization-time of flight (MALDI-TOF). Each of these techniques has their own specific advantages and limitations as to the types of data produced (Table 1). A critical advantage of MS is its sensitivity in analyte detection, where analytes can be precisely quantified rather than qualitatively compared. Additionally, the coupling of MS with separation techniques, such as GC or LC, offers considerable benefits to lipidome profiling by improving peak resolution and confidence in lipid species identification. While normal-phase LC (NPLC) or hydrophilic interaction LC (HILIC) methods can be utilized to obtain adequate separation of polar lipids such as glycerophospholipids or sphingolipids, reverse-phase LC (RPLC) capabilities enable enhanced separation of lipid species with predominantly hydrophobic lipids via interaction with 18-carbon side chains (C18) bound to silica beads.

**Table 1 metabolites-12-00333-t001:** Comparison of Mass Spectrometry Technologies.

Method	Advantages	Limitations	References
LC-QTOF/MS	High mass accuracy and resolution allowing for untargeted analyses and identification of unknown compounds. Can be used for structural elucidation of new lipid metabolites. High mass resolution.	Lower sensitivity than MRM mode scans, longer runs times, high cost.	[16]
LC-Orbitrap	Enhanced separation of isotopic peaks with similar retention times. High mass resolution.	Lower sensitivity than MRM mode scans, longer run times, high cost.	[17]
LC-Triple Quadrupole	NPLC, HILIC and RPLC capabilities, along with adjustable mobile phase gradient system enables optimized degree of separation between various lipid species. Enhanced sensitivity and selectivity of structurally similar lipids via multiple-reaction-monitoring (MRM) which utilizes precursor (Q1) and product ion (Q3) scans to differentiate lipid fingerprints. Identification and quantification of a large number of distinct lipid species. Reduced interference from biological matrices.	Non-uniformity across sample preparation, data acquisition, and data processing methods. Less effective for detecting unstable or reactive lipids (e.g., peroxides, radicals). Lower resolution than QTOF or Orbitrap.	[17,18,19,20,21,22,23,24]
Nano-ESI-MS	Small sample volume requirements. Steady ionization environment. High signal intensities due to low flow rates.	Longer run times, narrow needle can become clogged.	[25,26]
Acoustic Ejection MS	Used for liquid samples without need for sample extraction. Sample processing not required. Pulses of acoustic energy applied to the liquid surface to “lift up” a population of charged droplets. Offers high-speed, high throughput, and miniaturized experimentation.	Ion suppression of some analytes. May not be suitable for direct cell/tissue analysis.	[27,28]
GC-MS	Detection of volatile lipids such as free fatty acids and steroids.	Requires volatile analytes or derivatization to increase volatility.	[21]
MALDI-TOF	MALDI-TOF spectra can be used to generate 2-dimensional images depicting the localization of particular lipid species within a tissue sample. Can be employed after supplementation with isotope-labeled lipids (e.g., d2-AA or d2-DHA) to trace metabolic events and localization of metabolites. Reduced tissue preparation requirements.	No precursor ion selection or fragmentation leads to low confidence in identifying lipid species (unless MALDI-TOF/TOF is used). Lacks the resolution of QTOF and Orbitrap instruments.	[17,19,20]

### 2.2. Lipidome Stabilization

One limitation of “omic” approaches is analyte stabilization. Prior to analyzing any lipidomics sample, proper sample processing is imperative. There are several factors that affect the sample during lipidomic studies, especially the preprocessing, storage, selection conditions, and matrix specific sample processing [29]. Currently, there is a lack of information as to (1) the variability within normal healthy controls, (2) standardization of extraction procedures (e.g., recovery across concentration gradients), (3) evaluating the rate of standard and sample deterioration, (4) variation in tissue sampling, (5) differentiating between structural versus signaling lipids, (6) differences in data acquisition in relation to the types of equipment used, and (7) standardizing the bioinformatics principles applied.

For effective controls, it is important to know the age and condition of biological samples, as well as the extraction and preservation processes used. Since lipids are highly sensitive to oxidation, which can be introduced during sample preparation, the stabilization of susceptible analytes is of critical importance. Drying processed samples under nitrogen gas, as opposed to other techniques, is one approach to minimize the extent of lipid oxidation. Another important consideration is conducting stability studies which assess rates of deterioration. This is rarely evaluated, where known analytes are stored over time and the time-dependent degradation is evaluated after stabilization [30]. Using Arrhenius degradation curves, the original concentration of the analytes in the samples can be calculated. Most lipidomics techniques utilize biological tissues or extracts [31]. The most utilized lipid extraction systems are performed using a combination of organic solvents such as chloroform/methanol and water from a small biological sample or a combination of acetonitrile/methanol and butanol for larger samples [32]. Most organic solvents inactivate lipid metabolizing enzymes, which minimizes ex-vivo metabolism. Furthermore, sample collection into vials containing enzyme inhibitors may ensure sample integrity and lead to more reliable data. Some biological samples do not need to be processed or extracted such as for MALDI imaging, where typically the tissue is flash frozen to halt enzymatic activity and tissue sections are directly used. Sample processing and instrument variation are the cause for several errors throughout an analytical assay if data normalization techniques are not employed.

### 2.3. Data Validation

Data validation protocols should follow guidelines recognized by the FDA to expedite the drug development process. The Clinical and Laboratory Standards Institute (CLSI), a not-for-profit organization which develops global standards, has produced an FDA recognized guidance on evaluation of precise quantitative methods (EP05A3) [33]. To monitor and evaluate the quality control between lipidomic samples, an internal standard is added into all samples before processing. The internal standard is selected based on how structurally related it is to the lipid class of interest and should not be present in the blank matrix [34]. Recently, synthetic isotope-labelled lipid standards have been widely employed in lipidomics studies to reduce random systematic errors in detection. Their chemical similarities to endogenous lipids ensure adequate representation of extraction recovery levels. Isotopic labelled standards also have distinct chromatographic retention times and signals outside the natural mass distributions of endogenous analytes. Another important consideration, which was described previously, is the solvent used to reconstitute the internal standard to prevent unnecessary loss of sensitivity and prolong stability [35]. It is critical that the instrument and the analytical lipidomic assay are validated to ensure data accuracy, precision, and sensitivity across the assay range. Acquired data is processed for deisotoping, and normalization using the internal standard and can be quantified using a calibration curve [32]. The calibration curves for endogenous lipids raise a challenge due to the elevated background, however, multiple charcoal-stripped blank matrices are useful to address this issue. When the method is validated, the calibration curve should be reproducible, and validate that the matrix is free from potential interfering elements.

### 2.4. Data Analysis

The fundamental uniqueness of each biological system makes the analysis of metabolomic data more complicated despite shared biochemical reactions and metabolites [36]. Lipidomics can identify analytes against a large and complex background, where both experimental or environmental factors, such as genetics, ethnicity, age, diet, growth phase, biomatrices, nutrients, pH, sex, and temperature can play roles in metabolite concentrations [37,38]. The data output from lipidomics yields a complex matrix of variables that needs to be deconvoluted into fewer dimensions to spatially distinguish samples. Principal component analysis (PCA) reduces the dimensionality of the dataset into inferred variables, improving identification of major trends and features.

Lipidomic data dimension reduction may employ unsupervised methods, such as PCA, or supervised ones, like partial least squares regression (PLSR), PLS discriminant analysis (PLS-DA), and orthogonal projections to latent structures discriminant analysis (OPLS-DA). The primary goal of PCA and PLS is to identify group differences from a multivariate dataset. A class can refer to any biologically relevant classification, such as specific diet or drug interventions. PCA is the most widely used multivariate analysis method for feature-dependent sample classification. However, it is important for variables to be tested for the assumptions of predictive regression modeling before any meaningful conclusion can be extrapolated. These approaches are highly applicable to personalized medicine and therapeutic development.

More recently, machine learning algorithms have been employed to analyze data generated by newer bioanalytical techniques. Since metabolites can interact in a non-linear manner (e.g., enzymatic models), data structures from lipidomics data can become complex, making machine learning algorithms a practical approach to dissect data in a time-efficient manner [39]. These algorithms are commonly grouped as either (1) supervised, (2) unsupervised, or (3) reinforcement learning. Algorithm selection is important to balance data interpretation and accuracy. Choosing an algorithm is dependent on the outcome of interest, number of features and parameters, and training time for larger datasets. To expedite these types of modeling, two popular programming languages for machine learning are Python and R.

Following the development of predictive tools for neurological imaging, machine learning using Python has become an integral means for data analysis. Advantages of the Python language for lipidomics-based machine learning models include the wide variety of libraries available for data processing. NumPy and Pandas are data manipulation libraries built for mathematical computations, data cleansing, and data merging. The latter library can be further used to visualize and plot data along with Matplotlib and Seaborn libraries. Once data processing is complete, machine learning models can be built using Python libraries based on model and algorithm type (i.e., supervised or unsupervised, where supervised can be further broken down into classification vs. regression). Libraries such as sci-kitlearn (built upon SciPy) are open-source tools for predictive analyses and cover algorithms involved in both classification and regression such as support vector machine (SVM), k-nearest neighbors (KNN), and random forest. Furthermore, k-means, spectral clustering, feature selection, and cross validation algorithms are available for model selection and the optimization of features and parameters. These tools can be utilized to look at a fixed outcome of interest. Alternatively, deep learning using neural networks is a common example of unsupervised machine learning in which the algorithm is not provided with pre-defined labels or scores. Python libraries such as Keras (built upon Tensorflow) offer a neural network application programming interface (API) for which users can build layers and objectives into. These data analysis tools are useful in interpreting lipidomics data and presenting them in a form that can be used to extrapolate disease pathophysiology or for pharmacologic dissection.

## 3. Lipidomics in Disease Research and Pharmacology

Lipidomic analyses can be aimed at certain subgroups of lipid species such as steroids, triglycerides, phospholipids, polyunsaturated fatty acids (PUFA) and their bioactive metabolites (e.g., eicosanoids and docosanoids). Phospholipids are amphiphilic lipids characterized by a polar phosphate head group attached to two hydrophobic tails. These membrane-bound lipids are important structural components that support cellular membrane integrity and fluidity, but also serve as precursor molecules for various lipid subtypes by storing them in their esterified forms. Mammalian cells can contain thousands of distinct phospholipid species; however, the most abundant and widely characterized subgroups include the phosphatidylcholines (PC), phosphatidylethanolamines (PE), phosphatidylserines (PS), and sphingomyelins (SM) [40].

The hydrolysis of phospholipids and subsequent release of free PUFA into the cell is mediated by phospholipase enzymes, of which there are two major types. Calcium-dependent cytosolic phospholipase A2 (cPLA2) preferentially cleaves ω-6 arachidonic acid (AA) from phospholipids, whereas calcium-independent phospholipase A2 (iPLA2) more efficiently releases eicosapentaenoic acid (EPA), docosahexaenoic acid (DHA) and docosapentaenoic acid (DPA) ω-3 PUFAs. AA, EPA and their metabolites are termed eicosanoids, whereas DHA, DPA and their metabolites are termed docosanoids. Collectively, eicosanoids and docosanoids serve as potent regulators of inflammation both in systemic circulation and local microenvironments. These lipid metabolites exert their biological effects through binding to G-protein-coupled receptors (GPCR) found on the surface of various immune and glial cell types. Binding initiates signaling cascades to elicit cell-specific responses. Based on the type of immunologic response elicited, eicosanoids and docosanoids are classified as either pro-inflammatory or pro-resolving. While pro-inflammatory lipid mediators promote leukocyte recruitment, reactive oxygen species (ROS) generation and inflammatory cytokine release, pro-resolving lipids attenuate the inflammatory response by inducing leukocyte apoptosis and promoting the clearance of cellular debris by monocyte-mediated efferocytosis [41].

Most pro-inflammatory eicosanoids are products of AA metabolism (Figure 1). These include the prostaglandins (PGs) and thromboxanes (TXs), which are both converted from AA first by cyclooxygenase (COX) enzymes and subsequently by synthases. Alternatively, AA can be oxidized by 5-lipoxygenase (5-LOX) to form a pro-inflammatory intermediate, 5-hydroxyeicosatetraenoic acid (5-HETE), which can be further metabolized by 5-LOX to yield leukotrienes (LTs) [42]. In a similar manner, EPA can be oxidized by 5-LOX to generate 5-hydroxyeicosapentaenoic acid (5-HEPE) and LTs with inflammatory activities; however, EPA does not serve as a precursor for PGs or TXs [43]. Although 5-LOX mediated metabolism of AA and EPA lead to pro-inflammatory eicosanoids, oxidation of these PUFA by 15-LOX forms 15-HET(P)E and subsequently lipoxins (LXs) which are pro-resolving mediators capable of counteract the effects of LTs [44]. Moreover, 12-LOX activity facilitates the conversion of LTs to LXs. In addition to COX and LOX enzymes, AA can also be metabolized by cytochrome p450 (CYP) enzymes to generate epoxyeicosatrienoic acids (EETs), which are also pro-resolving lipid mediators, and in some cases, neuroprotective [45,46].

Unlike eicosanoids, the docosanoids are predominantly pro-resolving mediators (Figure 1). The 15-LOX mediated metabolism of DHA and DPA forms oxidized intermediates (HDHA and HDPA, respectively), which can undergo subsequent metabolism to generate D-series resolvins (RvDs) via 5-LOX or maresins (MaRs) and neuroprotectins (NPDs) via soluble epoxide hydrolase (sEH) [47]. These three classes of lipid mediators exert potent pro-resolving activities similar to LXs, while NPDs exhibit additional neuroprotective capabilities. In addition to the RvDs, there exists an EPA-derived group of resolvins termed the E-series resolvins (RvEs) [48]. These lipids have similar biological activities to their docosanoid counterparts; however, they have not been characterized as extensively.

Although hundreds of structurally diverse lipid mediators have been identified, their distinct physiological roles have only recently begun to be understood as technological advancements have enabled more comprehensive lipid profiling [44,47]. Since PUFAs and their metabolites have been shown to be important mediators of inflammation and pro-resolving activities, it is important to establish their roles in the initiation and progression of chronic inflammatory diseases. More importantly, understanding how lipidomic changes within a biological system mediate the transition back towards inflammatory homeostasis will open the door to effective therapeutic development. The ability to quantify these lipid mediators, accurately and precisely, will be critical for improving drug development strategies and monitoring the changes in inflammatory and pro-resolving lipids in response to therapy. However, pro-resolving lipids are biologically active at picogram concentrations and thus monitoring their changes at this level requires advanced instrumentation coupled with appropriate workflow, analysis, and controls [48]. The following sections will review how lipidomics has been utilized to understand various inflammation associated pathologies and their treatments including cardiovascular disease and stroke, neurodegenerative diseases, lung diseases, and autoimmune diseases. 

### 3.1. Cardiovascular Disease and Stroke

CVD is the leading cause of death regardless of gender or ethnic background [49]. It accounts for an estimated one third of worldwide deaths [50], encompassing coronary artery disease, myocardial infarction (MI), heart failure and stroke. Early diagnosis and prevention have been the focus of population-based health efforts to prevent CVD complications that result in mortality and long-term morbidity.

Aberrant levels of lipids and lipoproteins play an important role in the pathogenesis of CVD. Their association with atherosclerosis is well studied; however, lipids and lipoproteins also have central functions in cell signaling and the regulation of pro-inflammatory pathways [51]. In particular, oxidized cholesterols and low-density lipoproteins (LDL) are key biomarkers of diseases and are used to monitor responses to therapeutic interventions such as 3-hydroxy-3-methyl-glutaryl-coenzyme A reductase (HMGCR) inhibitors (e.g., simvastatin, atorvastatin). Elevation of these two key lipid-based biomarkers is central in the clinical risk assessment for CVD.

Cholesterols are both synthesized endogenously as well as gained from diet. Steroids like cholesterol are cellular membrane components that maintain the integrity and fluidity of cell membranes. Additionally, cholesterol is also a key precursor that forms bile acids and steroidal hormones. Both classes of steroidal structures are activators of nuclear factors (e.g., steroid xenobiotic receptor [SXR], liver X receptor [LXR], and constitutive androstane receptor [CAR]) and thus regulate metabolic enzymes such as the CYP system. In the blood, cholesterol, fatty acids (FAs), and triglycerides (TGs) are transported in hydrophilic lipoprotein complexes.

The contributions of cholesterols and lipoproteins to the development of atherosclerosis are well established. Although the development of atherosclerotic plaques is complex, dysregulation of lipid pathways plays a central role. Given the strong associations between atherosclerotic burden, ischemic cardiac disease, and stroke outcomes [52,53,54,55], a large focus of CVD management has been the use of lipidomic profiling of cholesterols and lipoproteins to predict ischemic CVD. Plasma biomarkers of CVD include high-density lipoprotein (HDL), LDLs, and TGs. Genetic, epidemiologic, and clinical lines of research have demonstrated that apolipoprotein B (ApoB)-containing lipoproteins, including LDL and very low-density lipoproteins (VLDL), are biologically associated with the development of atherosclerotic cardiovascular disease (ASCVD) through enhanced oxidation of LDL that promotes inflammatory signaling and cholesterol deposition in the vessel wall [56]. LDL profiling has been shown to be accurate and robust, where it is able to predict ASCVD in low and high-risk populations [57,58]. A comprehensive study of the bioactivity of lipids would improve the overall understanding of inflammatory mechanisms that underly CVD pathogenesis. In clinical practice, establishing novel biomarkers could aid in improving early diagnosis and identifying predictors of CVD outcomes. Lipidomics will aid in the characterization of CVD and therapeutic interventions.

#### 3.1.1. Lipoprotein Profiling of CVD

Plasma lipids are measured in several ways; however, LC-MS remains a popular research methodology because of its accuracy and high sensitivity [59]. While LC-MS techniques continue to evolve, density-gradient ultracentrifugation represents the gold standard for identification and quantification of LDL, HDL, and other sources of cholesterol esters (CEs) [60]. LC-MS cholesterol and lipoprotein analysis has been key to our understanding of lipid dysregulation underlying atherogenesis. In circulating blood, deficient conversion of free cholesterol to CEs, normally catalyzed by Lecithin-Cholesterol-Acyl-Transferase, is an early pathogenic step. In a lipidomic analysis of plasma donors with recent acute coronary syndrome or ischemic stroke, the ratio of CE to free cholesterol was lowest in CVD cohorts [61]. LC-MS can isolate specific metabolites or lipid compositions that are most atherogenic. LDL particles containing a CE-rich core of linoleic acid were less associated with carotid artery plaque formation [62]. Similarly, Stegemann et al. demonstrated that large polyunsaturated CEs showed the greatest relative enrichment within carotid plaques compared to plasma, and that a lipid signature of unstable plaques could be identified [63].

Future lipidomic applications will more accurately identify the cholesterol and lipoprotein metabolites that mediate ASCVD. If plasma LC-MS profiling can correlate unstable plaque lipid composition, this would establish a precise “liquid biopsy” of disease. A greater understanding of the biologic interactions between cholesterols and lipoproteins, and this how this transition leads to atherogenesis would provide insights as how to develop novel therapeutic interventions. The combination of LC-MS lipidomics in parallel with other approaches would allow a comprehensive “multi-omic” approach that potentially will yield significant scientific and clinical findings in CVD.

#### 3.1.2. Sphingolipids in CVD

Sphingolipids are a class of lipids that contain an aliphatic amino alcohol on the sphingoid backbone. Sphingolipids are either diet derived or produced through *de novo* biosynthesis. The complexity and variety of sphingolipids is representative of its diverse roles in the body. Like other lipid types, sphingolipids are structural lipids important in maintaining membrane integrity and function. In circulation, they help preserve lipoprotein structure and function. As signaling molecules, sphingolipids have been implicated in various essential physiological functions, such as growth inhibition, apoptosis, proliferation, differentiation, and inflammation [64].

The precise mechanistic role of sphingolipids in CVDs has not been fully elucidated, there is a growing body of evidence that supports the prognostic value of sphingolipids as predictive biomarkers for CVD outcomes including cardiovascular mortality, risk of heart failure, and recovery from MI [65,66,67]. Ceramide, a sphingolipid metabolite, is a secondary messenger regulating signals after tissue injuries. In chronic inflammatory conditions, ceramides exacerbate progression of atherosclerosis, where their presence has been noted in atherosclerotic plaques [68]. Ceramide accumulation in arterial plaques can activate transcription factors (e.g., nuclear factor kappa B [NF-κB], ets1 and PU.1) which further facilitate disease progression through uncoupling vascular nitric oxide signaling pathways, blocking insulin receptors signaling cascades, and inducing thrombus formation by platelet activation [69,70,71,72].

Aside from risk factor detection, circulatory sphingolipid profiles can change dynamically and are indicative of immunological response post-MI. The increases of different ceramide species have corresponded to upregulation of acute phase proteins and pro-inflammatory activation. Alterations in sphingolipid metabolism occurred during the reparative phase, 7 days after induced MI. In particular, ceramide-1-phosphate (Cer1p) to ceramide ratio in myocardial tissue corresponded with increased ceramide kinase (Cerk) expression [73]. As lipidomic profile studies investigate further into understanding its impact inflammation and tissue reparative process, lipid based diagnostic assays may become important in monitoring CVD outcomes.

The important role ceramides play in CVD progression also makes it an attractive target for treatment. Circulatory sphingolipid profiles and sphingolipid concentrations in plasma and in extracellular vesicles (EV) are known to fluctuate post-MI. Through the administration of the neutral sphingomyelinase inhibitor, GW4869, hydrolysis of sphingomyelins into ceramides is inhibited, which in turn impairs EV biogenesis. As a result, the suppression of circulating inflammatory EV post-MI in rats was able to preserve left ventricular ejection fraction [74]. This finding supports the hypothesis of inhibiting ceramide synthesis to prevent heart failure development post-MI. Ceramide inhibition was able to decrease ventricular remodeling, fibrosis, and inflammatory infiltrate [67,75,76,77]. There is significant prognostic and therapeutic value of ceramides and EV with regards to inflammation-driven CVD. Further elucidation as to the metabolism of sphingolipids may hold the key to managing chronic inflammation in CVD.

#### 3.1.3. Atherosclerotic Plaques

While there is an abundance of existing lipidomic studies that employ a comparative and systemic approach to elucidating the disease state of CVD at the serum and tissue level, there is limited research with the analytical resolution capable of quantitating CVD metabolites in single cells [78]. Atherosclerotic plaques are heterogenous deposits consisting of smooth muscle cells, mast cells, T-cells, B cells, myeloid cells, lipids, connective tissue, and other fibrous elements. Using single-cell transcriptomics and single-cell assays for transposase-accessible chromatin using sequencing (ATAC-Seq), Depuydt et al. [79] was able to identify distinct phenotypic subclasses within each immune cell population, highlighting the cellular plasticity and complex intercellular interactions at the disease site. In recent clinic trials for novel anti-inflammatory treatments against CVD, such as CANTOS, LoDoCo2, and CIRT [80,81,82,83] there is growing recognition that a standardized therapeutic approach is ineffective due to variation in patient responses. Optimization of therapies can be achieved by tailoring treatments targeting specific groups of CVD patients. This is predicated on their clinical and molecular biomarker status, including age, gender and genetic predispositions. Considering the heterogeneity of CVD plaques, a single-cell lipidomic analysis would expand our current understanding of the dynamic alterations and interactions between heterogenous cell subsets found in a plaque [84]. Ideally with this expanded understanding, we will be able to effectively personalize anti-inflammatory therapies.

#### 3.1.4. Stroke Research

Ischemic stroke is the most common form of stroke and a leading cause of death and disability worldwide [85]. Despite the burden of disease, beyond 24 h from the onset of acute ischemic stroke (AIS), no pharmacologic therapies exist beyond the secondary prevention of CVD and stroke [86,87]. Using lipidomics, a better understanding of mechanisms of secondary brain injury after stroke will lead to novel therapeutic approaches promoting neurologic recovery. Stroke pathology shares several genetic, behavioral, and epidemiologic risk factors with CVD. As previously described, the common pathologic pathway is atherogenesis mediated by inflammation and lipid dysregulation leading to ASCVD. Atherosclerosis in either intracranial or extracranial vessels is associated with stroke occurrence and post-stroke outcomes and is a target for early diagnosis and clinical risk stratification [54,55].

Lipidomic analysis of unstable carotid atheromatous plaques might provide a lipid signature to identify high-risk patients. Several lipid families associated with cell-signaling and inflammation have been studied. CEs are differentially associated with unstable plaques, with large polyunsaturated CEs showing the greatest vessel wall accumulation [62,63]. Profiling of PUFAs and their metabolites, including eicosanoids, offers additional differentiation. Levels of HETEs were significantly elevated relative to EETs in symptomatic patients compared to controls, with 9-HETE being the most abundant lipid measured [62]. Specialized pro-resolving lipid mediators (SPMs), particularly ω-3 PUFA-derived RvD1, have been shown to be significantly decreased in vulnerable plaque regions [88]. If unstable plaques could be identified by a “liquid biopsy”, early diagnosis and treatment would be possible.

Post-stroke lipidomics produces a unique metabolomic profile that might also differentiate stroke etiologies. LC-MS of serum samples after stroke demonstrated significant differences in FA metabolism of oleic acid, linoleic acid, and AA compared to healthy controls, while changes in phosphoglyceride metabolism was shown to differentiate small artery versus large artery occlusion [89]. Rodent models have been used to further elucidate dysregulation of lipid content following stroke. In a mouse middle cerebral artery occlusion (MCAO) model of AIS, Wang et al. demonstrated that changes in the post-stroke lipidome is most dynamic in the first 7 days and is characterized by an imbalance in phospholipid metabolism with reduced PC and increased lysophosphatidylcholine (LPC) levels [90]. In additional in vitro studies, the authors demonstrated that LPC reduced neuronal viability, while PC significantly suppressed microglial secretion of inflammatory cytokines IL-1β and (tumor necrosis factor alpha) TNF-α. Sphingolipid content is also altered after stroke. In a mouse AIS model, large increases in plasma ceramide and sphingomyelin were observed 24 h after stroke, with levels correlating with volume of ischemic brain tissue [91].

#### 3.1.5. Therapeutic Applications

Lipidomic studies in both human AIS and animal models of disease have identified potential therapeutic targets for stroke prevention and treatment. Growing interest has been in the administration of PUFAs and their anti-inflammatory metabolites. In a mouse model of ASCVD, oral supplementation of RvD1 restored RvD1: LTB4 ratios and reduced markers of oxidative stress and necrosis [88]. In a Sprague-Dawley MCAO model, intravenous administration of aspirin-triggered neuroprotectin D1 (AT-NPD1) 3 h post-stroke improved neurologic scores up to 7 days after stroke, reduced radiographic measures of cerebral edema, and decreased histopathologic infarct volume [92]. As recently reviewed by Miao et al. [93], several other SPMs, including LXs and maresins (MRs), have been studied as potential therapeutics after stroke. As a bioactive lipid family, emerging evidence suggests the important role SPMs play in attenuating inflammation after brain injury.

Antithrombotic drugs have been the main pharmacotherapy for CVD prevention. Among drugs that inhibit platelet activity, aspirin is the most well studied and most widely used therapeutic. In CVD, aspirins primary mechanism of action is the irreversible inhibition of platelet COX-1, which has been attributed to reducing TXA2 formation. Under normal conditions, TXA2 activates platelets and causes vasoconstriction through calcium-dependent pathways and promoting clot formation. Lipidomics has further elucidated the anti-inflammatory pharmacodynamics of aspirin. In addition to blocking endothelial cell PG synthesis by COX-1 and COX-2 inhibition, aspirin reduces pro-inflammatory oxylipins generated from PUFA oxygenation. In a study of healthy volunteers, low-dose aspirin was associated with a broad decrease in serum FA levels and reductions in linoleic acid and AA pro-inflammatory metabolites [94].

Aspirin can also promote resolution of inflammation by triggering the biosynthesis of SPMs by acetylation of COX-2. In AA metabolism, aspirin-treated COX-2 results in a 15-lipoxygenase (15-LOX)-like reaction that promotes the synthesis of aspirin-triggered lipoxins (ATLs) [95]. In experimental models, ATLs retain greater bioactivity as compared to endogenous LXs and inhibits neutrophil adhesion and transmigration across vascular endothelial cells [95,96]. The ω-3 FAs DHA and EPA are similarly metabolized by aspirin-acetylated COX-2. An increase in the formation of AT-E and D series Rvs has been demonstrated in several models of inflammation [95,97,98]. Although these findings have not been replicated in animal models of CVD, the clinical efficacy of aspirin may be partially explained by its promotion of anti-inflammatory pathways.

In CVD populations at the highest risk of cardioembolic disease, anticoagulant therapy is associated with reduced stroke risk. Among novel oral anticoagulants (NOACs), the direct thrombin inhibitor, dabigatran, has gained special interest for potential anti-inflammatory properties resulting from reduced thrombin activation of the PAR-1 receptor. Protease-activated receptors (PARs) are increasingly recognized for their role in mediating both acute and chronic inflammation. G protein-coupled signaling induced by PAR-1 activation promotes the expression of vascular cell adhesion molecule-1 (VCAM-1), intercellular adhesion molecule-1 (ICAM-1), and E-selectin, mediating vascular permeability to immune cells [99,100]. PAR-1 activation has been shown to mediate brain edema and neuronal cell death in a mouse model of global ischemia [101], identifying it as a potential therapeutic target in CVD. In a rodent sepsis model, dabigatran elicited soluble fibrinogen-like protein 2 triggered Rv-D5 production from DPA metabolism [102]. Dabigatran has also been shown to reduce lesion size, collagen content, and oxidative stress in hypercholesterolemic atherosclerosis [103]. Shotgun lipidomics may offer further insight into the action of dabigatran and other anticoagulants in mediating the inflammatory response after CVD.

Statins, or HMGCR inhibitors, are indicated to treat hypercholesterolemia and improve lipoprotein profiles in patients with CVD. Their anti-inflammatory and antioxidant effects have been well studied, including reduced levels of C-reactive protein, atherogenic LDL, TNF-α, interferon gamma (IFNγ), inhibition of T-helper cell inflammatory signaling, and by regulation of leukocyte-endothelial cell interactions [104,105,106,107]. Statins additionally promote the biosynthesis of pro-resolving SPMs. In a cardiac disease model, atorvastatin increased myocardial expression of 15-epi-lipoxin-A4 (ATLA) via S-nitrosylation of COX-2. Similar to aspirin-triggered acetylation of COX-2, S-nitrosylated COX-2 produces 15R-HETE, generating ATLA from 5-LOX activity [108].

Current research has additionally used a lipidomic approach to describe the differential effects of various HMGCR inhibitors and variabilities in patient response. In LC-MS plasma profiling of patients receiving statin therapy, rosuvastatin was associated with increased PC levels and lower sphingomyelin:(sphingomyelin + PCs) ratios when compared to atorvastatin [109], which may have clinical importance in the long-term management of CVD. Using samples from the Cholesterol and Pharmacogenetics study, lipidomic analysis was able to differentiate “good” from “poor” responders in subjects receiving simvastatin [110]. Together, these results highlight the power of lipidomics in the discovery of novel mechanisms of actions and patient-tailored therapies.

### 3.2. Neurodegenerative Diseases

Neuroinflammation is a common feature among many neurodegenerative diseases, such as AD, Parkinson’s disease (PD), and multiple sclerosis (MS). Considering that the brain is a lipid-rich environment where lipid uptake and metabolism is tightly regulated, lipidomics-based approaches are becoming more widely recognized as effective strategies to investigate neurodegenerative disease pathophysiology and identify clinical biomarkers. This section describes the interplay between lipid mediators of inflammation and hallmarks of neurodegenerative disease pathologies, how these interactions may influence disease progression, and how lipidomic profiling has been utilized to evaluate the efficacy of pharmacological interventions against neuroinflammation and neurodegeneration.

#### 3.2.1. Alzheimer’s Disease

AD progression has been closely linked to the accumulation of beta amyloid (Aβ) peptides and neurofibrillary tangles for several decades, yet therapeutics developed to target these aggregates have not corresponded to effective therapeutics. More recently, chronic neuroinflammation has become recognized as a central hallmark of AD pathology, where glial cell activation (e.g., microglia and astrocytes) results in the release of pro-inflammatory cytokines and eicosanoids to promote neurodegeneration [111].

Of the eicosanoids generated from glial cell activation, LTs and PGs have been the most widely studied in AD pathogenesis. They have been detected at higher levels in the AD brain and implicated as mediators of AD-related neuroinflammation [112]. LTB4 has been linked to increase Aβ production and chronic gliosis [112,113,114], while PGE2 is largely produced by activated microglia and is known to induce AD-like phenotypes in astrocytes [115,116,117]. 5-LOX, which facilitates the biosynthesis of LTs and several other eicosanoids, has gained interest as a potential drug target against neuroinflammation because it is widely expressed in the central nervous system (CNS) and co-localizes with Aβ and neurofibrillary tangles in the hippocampal region of AD patients [118]. These findings suggest that 5-LOX activation and its downstream metabolites such as LTs may contribute to plaque deposition in AD, although further clinical studies are warranted to clearly define the relationship.

Neurotoxic aldehydes generated from PUFA peroxidation such as 4-hydroxy-2-nonenal (4-HNE) and malondialdehyde (MDA) have been linked to AD pathogenesis [119]. These reactive lipid species are readily formed in environments plagued by chronic oxidative stress, such as hippocampal and cortical regions of the AD brain. In clinical cohorts, 4-HNE levels have been positively correlated with cognitive decline and AD severity [120,121]. While these lipid peroxidation products are highly unstable and therefore difficult to detect using traditional MS methods, many procedures have been developed to accurately quantify their levels by converting 4-HNE and MDA to readily detectable products using derivatizing agents [122,123].

The ω-3 PUFAs (i.e., EPA, DPA, DHA) and their metabolites are recognized as protective in the context of AD. These metabolites include the Rvs, MaRs, and NPDs, the majority of which are derived from DHA, which constitutes up to 60% of all esterified fatty acids in neuronal plasma membranes [124]. In clinical cohorts, DHA and NPD1 deficiencies have been associated with cognitive decline [125]. NPD1 not only possesses anti-inflammatory capabilities, but also exerts neuroprotective effects by promoting neurotrophic cell survival and suppressing amyloidogenesis via inhibition of Aβ peptides [124]. As such, lipidomic studies comparing hippocampal regions between AD patients and age-matched controls found DHA and NPD1 to be significantly less abundant in brain tissue from AD patients. Interestingly, this trend has only been documented in brain regions which are affected by AD pathology (e.g., cortex, hippocampus) and has not been demonstrated in other compartments of the brain such as the thalamus [125,126,127].

In clinical AD studies, the ratio of ω-3:ω-6 PUFA is often measured to assess the degree of inflammation, where a lower ratio is indicative of a more proinflammatory state [128,129]. This is attributed to the fact that AA (ω-6) is the precursor to the majority of pro-inflammatory PUFA derivatives such as LTs, PGs, and TXs, whereas ω-3 PUFAs are largely metabolized into pro-resolving and neuroprotective lipid mediators. cPLA2, which exhibits a specificity toward AA-containing phospholipids, has been linked to glial cell mediated neuroinflammation by promoting the release of free AA and inducing ROS formation via the LOX pathway [130]. Postmortem analysis of human cortical samples revealed higher cPLA2 activation in AD brains with the apolipoprotein E ε4 allele (APOE ε4), which is associated with more severe patient outcomes and neuroinflammation [131,132]. These findings suggest that in addition to lipid transport, APOE genotype may also play a significant role in regulating lipid metabolism through cPLA2. Lipidomic profiling of brain tissues from AD patients could help clarify the mechanisms driving these relationships, although large sample sizes would be necessary to overcome the high degree of variability between AD cases.

#### 3.2.2. Parkinson’s Disease

Similar to AD, PD is a neurodegenerative disorder characterized by neuroinflammation and oxidative stress in the brain, which is driven by chronic glial cell activation [133]. This is evidenced by increased TNF-α and NF-κB activation in brains with PD, along with increased production of pro-inflammatory cytokines and ROS [134,135,136]. Additionally, dysregulation of PUFA metabolism has been implicated in PD pathophysiology, and several pro-inflammatory lipid mediators have been identified as potential biomarkers for the disease.

For example, increased levels of PGD2 and PDE2 in brains with PD have been detected, which has coincided with increased COX-2, prostaglandin E synthase and prostaglandin D synthase activities [137,138,139]. In a neurotoxin-induced mouse model for PD, prostaglandin E synthase activity and PGE2 production were shown to coincide with brain lesions and dopaminergic neuronal death [139]. In addition to prostaglandins, markers of lipid oxidation such as 4-HNE and HETEs have been shown to be elevated in PD patients when compared to controls, which contributes to neurodegeneration by promoting inflammation and oxidative stress [133,140,141]. The role of SPMs in PD has not yet been widely investigated; however, a recent clinical study comparing PD patients with healthy age-matched controls found significantly lower levels of RvD1 in both the CSF and plasma of individuals with PD [142,143]. Further lipidomic studies of this type are necessary to construct a more complete understanding of the PD lipidome, which could delineate relationships between lipid metabolism and disease progression, thus guiding drug development strategies.

Lewy bodies, which are protein aggregates comprised of α-synuclein, represent an important neuropathological hallmark of PD. Structurally, α-synuclein is very similar to A2 lipoproteins such as cPLA2, and it has been shown to be involved in phospholipid metabolism [144,145]. Both cPLA2 and α-synuclein have been shown to have increased activity in the PD brain [133]. Recently, α-synuclein accumulation and propagation has been shown to be highly dependent on the types of lipids which α-synuclein binds to. DHA for example, which makes up >50% of esterified PUFA in plasma membranes, has been shown to increase α-synuclein expression and aggregation [144].

#### 3.2.3. Therapeutic Applications

Lipidomic profiling can support the development of therapeutics for neurodegenerative diseases by identifying enzymatic drug targets responsible for glial cell class switching toward pathological phenotypes. Additionally, precise and accurate lipidomics can be used to monitor efficacy of potential drug candidates. As our understanding of lipid mediators and their involvement in neurodegenerative diseases continues to unfold, many have investigated different approaches to regulate neuroinflammation by PUFA supplementation or targeting proteins within the PUFA metabolism pathways such as PLAs, COXs, and LOXs.

Due to clinical evidence showing lower levels of DHA and its metabolites in AD patient brains compared to age-matched controls, supplementation with DHA and other ⍵-3 PUFAs has been widely explored as a possible treatment or preventative action against AD-associated cognitive decline. Although dietary DHA supplementation in transgenic mouse models has been shown to mitigate Aꞵ pathology and restore cerebral blood volume, DHA supplementation in humans has yielded inconclusive results. Meta-analyses of several observational studies and clinical trials did not find conclusive evidence of improved cognitive function or protection against AD pathologies after short, medium, or long term (>6 months) daily DHA supplementation [146,147,148]. Some have hypothesized that these clinical failures stem from larger doses of DHA being required for adequate brain bioavailability, however, a recent study by Arellanes et al. showed that after long-term daily supplementation of 2,152 mg DHA, changes in brain volume and cognitive function were no different from the placebo group despite a 28% increase of cerebrospinal fluid (CSF) DHA [149]. Employing lipidomic strategies such as targeted LC-MS/MS and MALDI-TOF imaging may provide insights as to how DHA supplementation has failed in humans; these approaches can establish the relationship between DHA levels in the plasma, CSF, and brain, as well as determine DHA metabolites localization in the brain.

The overactivation of cPLA2 in brains of patients with AD and PD, and its selective induction of the pro-inflammatory AA pathway, has led many to hypothesize that cPLA2 inhibition may ameliorate neurodegenerative disease outcomes by shifting lipid metabolism toward the production of pro-resolving mediators derived from ω-3 PUFAs. Inhibition of cPLA2 by Annexin A1 has been shown to elicit neuroprotective effects by mitigating neuroinflammation and neuronal damage in the CNS following spinal cord injury to rats, and cPLA2 knockout in AD mouse models was shown to ameliorate cognitive deficits despite Aꞵ accumulation [150,151]. In a GH3 dopaminergic neuron cell line for PD, cPLA2 inhibition by arachidonyl trifluoromethyl ketone was shown to reduce MPTP(1-methyl-4-phenyl-1,2,3,6-tetrahydropyridine)-induced cytotoxicity; however, cPLA2 inhibitors have not been thoroughly investigated in PD animal models [152]. As cPLA2 inhibitors have yet to be widely assessed in humans or animal models for other neurodegenerative diseases, lipidomic profiling can serve as a comprehensive analytical tool to evaluate the efficacy of cPLA2 inhibitors and support developmental efforts.

Aspirin, a COX inhibitor, has been widely explored as a potential treatment for protection against neurodegenerative diseases such as AD because of its ability to block PG biosynthesis and promote the formation of ATLAs. A meta-analysis of 12 cohort studies and three clinical trials reported that although low doses of aspirin coincided with reduced incidence of dementia in cohort studies, these findings were not confirmed in clinical trials [153]. Similarly, the COX inhibitors ibuprofen and indomethacin yielded inconclusive results in AD clinical trials, suggesting that COX inhibition alone may be insufficient for preventing cognitive decline [154,155]. In a lipidomic study of healthy individuals receiving low daily doses of aspirin, notable inter-subject variability in systemic eicosanoid levels were reported, and unexpected reductions in non-COX mediated eicosanoids were also observed [94]. These findings could partially explain why COX inhibitors have failed in clinical trials. To better understand the effects of aspirin in neurodegenerative diseases, similar large-scale lipidomic profiling studies should be conducted in elderly populations where CSF eicosanoid levels can be quantified. The utility of specific COX-2 inhibitors in PD has also been investigated in preclinical studies, where COX-2 inhibition in PD rodent models correlated with less brain lesions and increased protection of dopaminergic neurons [156]. Still, these studies have not been tested in humans because the mechanism by which COX-2 activity promotes neurodegeneration in PD rodent models remains unclear.

In addition to COX enzymes, the inhibition of LOXs have been proposed for the treatment or prevention of neurodegenerative diseases such as AD. The inhibition of 5-LOX prevents LT formation from AA, and in an AD mouse model, 5-LOX inhibition has been shown to reduce cognitive impairment, Aꞵ deposition, and neuronal loss [157]. Furthermore, 5-LOX inhibitors have been shown to reduce microglia-mediated toxicity towards neuronal cells in human cell lines [157,158], and similar effects were observed with the selective inhibition of leukotriene receptors [159]. Although COX inhibitors alone have been ineffective against AD in clinical trials, the combination of COX and 5-LOX inhibitors has proven to be more effective in mitigating microglia-associated toxicity compared to single inhibitors [160], and thus may require additional analysis to affirm these findings.

In the quest to identify anti-inflammatory drugs for AD and PD, minocycline has been tested in clinical trials because of its ability to suppress oxygen radical formation and microglial activation [161]. The drug, however, failed in clinical trials for AD due to lack of efficacy and the occurrence of adverse effects which could be attributed to several reasons. For one, minocycline targets and anti-inflammatory mechanism of action are not clearly defined. Second, the adequate dose of minocycline required for suppression of reactive lipid species has not been determined [162,163]. To address these questions or prevent similar failures in future drug development, lipidomic profiling in relations to drug concentrations is required to better understand the pharmacologic parameters required for optimal pharmacodynamic activities.

### 3.3. Inflammatory Lung Diseases

Inflammation is a common pathological feature for several lung diseases. There is growing evidence supporting the role of lipids in mediating inflammation of bacterial and viral infections, chronic obstructive pulmonary disease (COPD), asthma, cystic fibrosis, and cancer [164,165,166,167,168,169,170,171,172,173,174,175]. Lipidomics can provide insights into pathophysiology, drug toxicology, drug mechanisms, and clinical biomarkers in lung disease. This section will review lipidomic findings from various lung diseases and address how lipidomics is being utilized to drive drug development for treatment of lung inflammation.

#### 3.3.1. Biomarker Identification

COPD is one of the leading causes of mortality in the US and worldwide and lacks effective treatments [176]. Additionally, there is a lack of predictive biomarkers for COPD outcomes which further complicates the drug development process. Using immunoassays, it was shown that 8-isoprostane, LTB_4_, and PGE_2_ were significantly increased in the sputum of COPD patients [177]. Lipidomics can provide the robustness required to establish these lipid mediators as clinical biomarkers. Lipidomics combines excellent sensitivity with improved sample workflow to provide high quality data on a higher number of analytes. In addition, LC-MS lipidomics require smaller sample volumes with a faster processing time. Lipidomics has revealed alterations in the blood lipid profiles of COPD patients and identified species of phospholipids, glycerolipids, sphingolipids, and sterol lipids as potential biomarkers of disease [178]. However, data validation will require a larger study and a targeted lipidomic approach. A large portion of lipidomic studies have focused on sphingolipids and phospholipids, however targeted lipidomics focusing on PUFA and their metabolites may reveal novel therapeutic targets.

Lipidomic studies have identified critical changes of the lipidome in Severe Acute Respiratory Syndrome Coronavirus 2 (SARS-CoV-2) in relation to disease severity [179]. Globally, there are over 400 million cases with 5 million deaths associated with this infection [180]. Progression of SARS-CoV-2 infection can lead to respiratory failure, and acute respiratory distress syndrome (ARDS) characterized by acute lung injury. Risk factors of severe infection associated with pathological lipidome changes include age, hypertension, diabetes, and obesity [181,182,183,184].

In particular, dysregulation of eicosanoids and docosanoids in the serum and lungs from SARS-CoV2 patients showed changes in sphingomyelins, phospholipids, glycerolipids, and CEs which have been associated with other lung injuries and cardiovascular diseases as described previously [179]. Of specific interest, moderate disease patients had significantly increased levels of pro-resolving RvE3 and trending increased levels of PGs. Moderate disease was associated with higher levels of COX activity and EPA metabolites of 12-LOX. In contrast, severe SARS-CoV2 was associated with increased 5-LOX and CYP activities. As expected, SARS-CoV2 patients had increased AA and its metabolites. SPMs including RvDs were also increased in bronchoalveolar lavage (BAL) of SARS-CoV2 patients. In agreement with these findings, Zaid et al. found high concentrations of multiple cytokines, chemokines, and lipid mediators in BAL from severe SARS-CoV2 patients [185]. In a targeted analysis, PGs, TXs, and LTs were found in higher concentrations and contributed to inflammation and neutrophil influx [186]. Archamabult et al. showed SARS-CoV-2 infected patients requiring intubation have dysregulated pulmonary levels of eicosanoids and docosanoids [174]. Unfortunately, blood marker analyses did not correlate with BAL findings, indicating regional lipid dynamics in lung injury and inflammation. Larger studies are required to establish lipidome profiles of healthy and patient populations and to enable lipidomic investigations of drug mechanisms and efficacy.

#### 3.3.2. Lung Toxicology

Lipidomics can also be used to monitor toxicological responses of therapeutic interventions. For example, it has been used to evaluate bleomycin, an anticancer agent, induced lung toxicity in a mouse model [187]. In the acute inflammatory phase following treatment, AA metabolites (e.g., PGD_2_ and PGE_2_) were increased while DHA metabolites increased on day 7 during the inflammatory-to-fibrosis phase. Interestingly, no plasma lipidome changes were observed with bleomycin treatment, indicating the need for appropriate sample collection to properly study drug-induced lipidome alterations. LOX inhibition using nordihydroguaiaretic acid has been shown to attenuate bleomycin induced lung fibrosis, which suggests that bleomycin may exert its pulmonary toxic activity through dysregulating lipid metabolism [188]. Utilizing lipidomics could provide insight as to which LOX metabolites are critical for preventing drug induced lung fibrosis.

Lung injury is also a common adverse effect of radiation therapy where no approved treatment or medical countermeasures for radiation-induced lung injury (RILI) exist. Decreased pulmonary surfactant lipids (i.e., PC series) and heme b were evident following RILI in rhesus macaques regardless of pathological presentation [167]. Additionally, RILI was associated with decreased sphingomyelins and increased PUFAs which can be linked to pro-inflammatory and pro-resolution mechanisms as described previously.

#### 3.3.3. Therapeutic Applications

It is well established that lipid profiles are altered in respiratory infections due to an increase in COX-2 expression. COX inhibitors such as non-steroidal anti-inflammatory drugs (NSAIDs) are routinely used to treat inflammation and pain associated with a myriad of conditions [175]. Several classes of drugs have been used to treat SARS-CoV-2 including steroids, COX inhibitors, and antivirals. Aspirin treatment was able to reduce the need for mechanical ventilation and ICU admission rates [189]. In animal studies, COX, neuraminidase, and 5-LOX inhibitors have been shown to decrease levels of inflammatory cytokines [190,191]. Remdesivir (GS-5734), an antiviral inhibitor of viral RNA polymerase, was first approved in 2020 for the treatment of SARS-CoV2 [192]. Du et al. found that rat plasma lipidomic profiles showed DHA, RvD2, 5-HEPE, and 5-HETE levels decreased following remdesivir treatment while TXB2 increased [193]. This study showed reduction in the lipoxygenase pathway metabolites may delay inflammation resolution. However, due to a lack of assessment in a disease model, more research is required to understand the impact of remdesivir on lipidome profiles of COVID-19 patients.

Pharmacologic intervention such as Ramatroban (Baynas^®^, Bayer, Tokyo, Japan), a dual receptor antagonist of D-prostanoid receptor 2 (DPr2) and TX receptors (TDRs) has been used in Japan to boost interferon lambda (IFN-λ), thereby suppressing SARS-CoV-2 replication. In a small cohort of patients, 75 mg twice daily of Ramatroban administration was found to rapidly improve both respiratory distress and hypoxemia. Reduction in disease severity was able to prevent hospitalization and promote recovery from acute disease [194,195]. Although the proposed pathways are thought to inhibit DPr2 and TDR mediated activities, it is still not clear if Ramatroban will have effects on additional lipid metabolism.

Glucocorticoid receptor agonist, dexamethasone, was shown to reduce mortality in SARS-CoV2 patients [196]. Priyillou et al. has shown that dexamethasone induces the D-series pro-resolving lipid mediator pathway, which may explain the ability of dexamethasone to induce eicosanoid class switching to pro-resolution [197]. It is hypothesized that AA-derived EETS could attenuate SARS-CoV2-induced hyperinflammation. It has been proposed that sEH inhibitors could be utilized in SARS-CoV2 (e.g., GSK-2256294) and other respiratory infections [198,199]. Eicosanoid signaling is tied to immune function, and therefore further research is critical to understand the basis of various drug classes effects on healthy and pathological lipidome profiles.

### 3.4. Autoimmune Diseases

Autoimmune diseases are characterized by chronic inflammation and loss of immune tolerance. If not treated early and effectively, disease progression towards chronic systemic inflammation is likely. Production of self-reactive autoantibodies and tissue injury results in lipid release into circulation, altering the plasma lipidome and indicating active chronic inflammation. Systemic Lupus Erythematosus (SLE) and irritable bowel disease (IBD) are some of the most persistent autoimmune diseases that have no highly specific diagnostic tests available, impairing treatment and recovery times. To overcome this, lipidomics was identified as a novel approach to provide more insights into autoimmune disorders and to identify potential lipid biomarkers which would further help characterize and understand the underlying disease pathophysiology.

#### 3.4.1. Systemic Lupus Erythematosus

SLE is a chronic autoimmune disorder found predominantly in women which can affect several organs in the body including skin, kidney, liver, and brain. The underlying molecular mechanisms of this disease are largely unknown; however, SLE pathogenesis is attributed to oxidative stress and dysfunction of the immune system. ROS contributes to dyslipidemia and dyslipoproteinemia which are characterized by high levels of VLDL and TG but lower levels of HDL. Dyslipidemia occurs because of declining renal function, which is one of the most common comorbidities for SLE. Therefore, SLE patients also have a higher susceptibility towards developing CVD. HDL promotes oxidation of LDL and is a cell efflux promoter, which can promote NETosis and compromise lipid metabolism when dysregulated [200]. SLE patients have periods of disease flares and remission which greatly affect the plasma lipid profile. Determining lipidomic profiles at molecular levels could lead to new detection strategies and biomarker discovery. GC-MS and multi-dimensional shotgun MS of lipid profiles evaluated FAs, free FAs (FFA), sphingolipids, TG, and PL levels in SLE patients. The analysis revealed that TGs were increased, and levels of lysophosphatidylethanolamine (LPE) and PC significantly decreased in patients diagnosed with SLE. These conclusions were attributed to three major pathways that included decrease of cPLA2 activation along with peroxisomal dysfunction and degradation. Additionally, it was found that under oxidative stress, plasmenyl-PE species have an antioxidant role which supports normal cellular functions. Therefore, determining plasmenyl-PE levels could be used as a potential biomarker to determine SLE prognosis and ROS [200]. FA composition in circulation can change with diet and medication use and have thus been associated with improving SLE symptoms in most patients.

Oleic acid, EPA, and AA have also been measured at lower levels in SLE patients, where reductions of these lipids were associated with higher disease severity. Early stages of autoimmune diseases showed PGs and LTs induce influx of neutrophils and increase biosynthesis of PGE2 and LTB4 [201]. Selectively inhibiting cPLA_2_ resulted in reduction of PGE2 and LTB4. However, chronic use of corticosteroids (e.g., dexamethasone) could potentially lead to a flare-up and failure to treat the inflammatory disease state [202]. This effect may be attributed to inhibition of cPLA2 which is upstream of both inflammatory and pro-resolving lipid mediators (Figure 1). Increased dietary fish oil containing DHA and EPA can shift SLE patient lipid profiles by providing the precursors to make SPMs. In a study conducted in two toll-like receptor (TLR)-7 agonist induced lupus mouse models, EPA treatment suppressed autoantibody production and opsonization complex depositions in the glomerulus. In addition, EPA suppressed B-cell differentiation which would in-turn prevent autoantibody production [203]. The practical use of lipidomics analysis is the identification of overlapping lipids that were identified via shotgun MS among multiple autoimmune diseases. Overlapping lipid profiles between multiple pathologies suggests the existence of global biomarkers for autoimmune diseases. This area of research is in the beginning phases and shows potential for developing the next generation of autoimmune therapies.

#### 3.4.2. GI Autoimmune Diseases

Crohn’s disease (CD) and ulcerative colitis (UC) are IBD subtypes [204]. While both cause chronic inflammation of the gastrointestinal tract, their etiology and pathological mechanisms are not clearly known. Currently, diagnostic approaches rely on clinical manifestations, such as endoscopic, histological, and radiological findings [205]. However, there is no standard diagnostic tool for IBD involving specific biochemical biomarkers. An untargeted exploration of plasma samples of UC patients showed the precursors of Rvs, EPA and DHA, significantly increased in UC patients with disease stage [206]. In biopsy samples of UC patients, inflamed mucosa showed higher AA, lower EPA, and a higher AA:EPA ratio. Inflamed mucosa revealed higher levels of DPA and DHA, and lower linoleic acid and α-linolenic acid levels compared to non-inflamed and healthy controls [207]. The inflamed mucosa also showed significantly higher levels of PGE2, PGD2, TXB2, 5-HETE, 11-HETE, 12-HETE, and 15-HETE which correlated with severity of inflammation.

SPMs effects on intestinal inflammation include reduction of NF-ĸB activation, decreased neutrophil infiltration, and the phenotypic switch of macrophages from pro-inflammatory to pro-resolving [208,209,210,211,212]. Consistently, 15-LOX inhibitor (e.g., PD146176) administration inhibited SPM production leading to worsening of colitis in mice. Inversely, aspirin administration increased LXA4 and 17-hydroxy DHA and decreased colon inflammation [212,213,214]. Impaired LX biosynthesis was found in colonic mucosa from UC patients while lipoxin A4 (LXA4) levels were increased in the mucosa of those in disease remission, along with increased macrophage infiltration and increased mRNA of LXA4 receptor formyl peptide receptor 2 (FPR2/ALX) [213,215]. LXA4 levels were also found to be negatively correlated with histopathologic alterations in experimental colitis, suggesting a beneficial role for this SPM in IBD [216]. NPD1 levels were increased in the colon of mice with dextran sulfate sodium (DSS)-induced colitis and decreased in colonic tissue of eosinophil-deficient mice that develop more severe acute colitis compared to control mice [209,212]. Accordingly, administration of an exogenous NPD1-isomer ((10S,17S)-DiHDoHE) reduced neutrophil infiltration and inflammatory markers, thus reducing DSS-induced colitis severity in eosinophil-deficient mice [209]. The EPA-derived RvE1, aspirin-triggered resolvin D1 (AT-RvD1), its precursor (17R)-hydroxydocosahexaenoic acid, RvD2, and Maresin1 have shown beneficial effects in acute and chronic experimental colitis [208,211,217,218,219].

Abnormal FA metabolism has been shown in IBD patients with both active and quiescent states. In particular, PUFA dysregulation is found in the bowel inflammation process through eicosanoids derived from AA corresponding to increased colonic inflammatory cytokines and increased serum FA [220,221,222]. In this study, fatty acyls were shown to be the most significantly disturbed lipid species in IBD patients. As a representative of PUFAs, the metabolism of AA exerts a pivotal function in the inflammatory response including formation of inflammatory factors and ROS generation. The AA metabolite 20-HETE is catalyzed from CYP enzymes and regulates inflammatory vascular response through its interaction with nitric oxide [223]. Similarly, epoxy-eicosatrienoic acid (EpETrE) is a derivative of AA and plays a role in mediating the effects of inflammation on blood vessels [224]. AA metabolic pathway in IBD has dysregulated PGE2 corresponding with reduction of EPA and its metabolites [225]. Decreased EPA, DHA, AA, 20-HETE, (+/−)5,6-EpETrE, and increased (+/−)8,9-EpETrE were found in IBD patients. Similarly, UC patients exhibit alterations in the AA/EPA ratio, and the amounts of AA, DPA, DHA, LA, α-LNA, and EPA are associated with the severity of inflammation [207]. These findings suggest lipidomics approaches can detect and monitor IBD. However, more robust studies of intervention associated alterations in lipidome profiles need to be conducted to confirm the hypothesis of targeting pro-inflammatory lipid mediators in the treatment of GI pathologies.

## 4. Limitations

This review is meant to give readers an understanding of lipidomic methodologies and considerations with examples of how lipidomics can drive pathophysiology and pharmacologic dissections. However, due to the vast amount of literature, the review aims to be concise to capture the multitude of factors involved in lipidomic studies but is not exhaustive. Additionally, this review is limited to focus on selected inflammatory diseases, while there are several other inflammatory diseases showing dysregulation of the eicosanoid pathways that were not addressed. Another limitation concerns methods both for data production and treatment. Sample preparation, experimental conditions, and analytical methods were highly variable between studies, which made it difficult to draw comparisons between them. For example, different lipid stabilization methods, storage times, and extraction techniques were utilized. Additionally, this review does not address isobaric or isomeric lipids which may coelute together. The studies reviewed were conducted with different MS technologies and across different biomatrices, which made it difficult to form uniform assessments. None of the studies evaluated address the issues of comprehensive medication history for study participants, and thus drug utilization impact on lipid concentrations cannot be delineated. Lastly, the review does not address other metabolomic or proteomic mediators that facilitate the lipidomic signaling process. Metabolites involved in the generation of ROS work closely with the inflammatory lipids but have been excluded from the scope of this review. A summary of the lipidomic studies discussed in this review can be found in Table 2.

## 5. Conclusions

Lipidomic profiling has advanced our understanding of several diseases. Although specific analytes and their changes in concentration have been important to identify potential dysregulation, the ratio of the precursors in relation to bioactive metabolites may be more informative. This highlights the delicate balance between inflammatory and pro-resolving mediators, but also the importance of enzymes involved in lipid metabolism pathways. In the lipidomic profiling of the various diseases discussed in this review, active inflammation is associated with lower EPA:AA and DPA:AA ratios in the early onset of acute inflammation, where a counterbalance with pro-resolving lipid mediators can promote disease resolution. In chronic inflammation, however, persistent lowering of EPA:AA and DPA:AA ratios and the inability to realign this metabolic imbalance is apparent. There are efforts utilizing this data to develop therapeutic strategies; where dysregulation in the lipidome may also present an opportunity to develop precise biomarkers for disease monitoring. These opportunities can only be realized when the affected tissues have lipidome changes that can be detected in the circulation or affected tissues can be readily sampled as in BAL lung samples. In order to do these types of correlations, machine learning and deeper learning tools are required to improve their predictive values. It is key that advancement of analytical techniques is integrated with machine learning analysis correlating clinical outcomes with these biochemical markers. Standardization of lipidomic analyses will be critical in the drug development process to ensure accurate and reproducible findings from various academic and industry laboratories. Researchers interested in utilizing lipidomics should refer to agency recognized guidelines on assay development.

This review highlighted several considerations for lipidomic analyses including advantages of various MS instruments, sample stabilization, and data analysis and validation. Furthermore, this article reviewed how lipidomics has revealed novel biomarkers of inflammatory diseases and driven drug development by elucidating molecular mechanisms of various pharmaceutical agents. Lipidomic findings have shown strong overlap among these inflammatory diseases and allow researchers to utilize analyses in other systems to drive their own research and development. More comprehensive studies in the future can result in larger data-banking and, with the use of advanced computing, accelerate development of novel therapeutic interventions in inflammatory diseases on a global scale.

## Figures and Tables

**Figure 1 metabolites-12-00333-f001:**
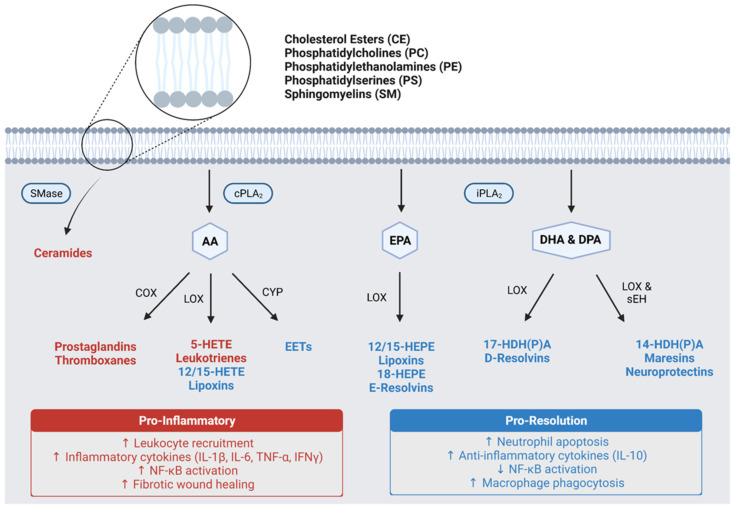
Pro-inflammatory and pro-resolution lipid metabolism pathways. AA, arachidonic acid; EPA, eicosapentaenoic acid; DHA, docosahexaenoic acid; DPA, docosapentaenoic acid; cPLA2, calcium-dependent phospholipase A2; iPLA2, calcium-independent phospholipase A2; SMase, sphingomyelinase; COX, cyclooxygenases; LOX, lipoxygenases; CYP, cytochrome P450; EETs, epoxyeicosatrienoic acids; HETE, hydroxyeicosatetraenoic acid; HEPE, hydroxyeicosapentaenoic acid; HDHA, hydroxydocosahexaenoic acid; HDPA, hydroxydocosapentaenoic acid; NF-ĸB, nuclear factor-kappa B; IL-1β, interleukin-1 beta; IL-6, interleukin-6; TNF-α, tumor necrosis factor-alpha; IFNγ, interferon gamma; IL-10, interleukin-10.

**Table 2 metabolites-12-00333-t002:** Summary of lipidomics studies in inflammatory disease research.

Disease/Injury	Pharmacologic Agent	Mass Detection	Lipid Source	Findings	References
CVD	None	Orbitrap	Human Plasma	The ratio of CE to free cholesterol is lowered in CVD patients	[61]
	None	LC-triple quadrupole, Shotgun MS	Patient tissue sections/extracts, plasma	Polyunsaturated CE are largely enriched in carotid plaques	[63]
	None	LC- triple quadrupole	Mouse heart tissue, plasma	Upregulated drastically in tissue after myocardial injury to activate cellular regeneration and inhibit pro-inflammatory cytokines	[73,74]
	RvD1	LC- triple quadrupole	Mouse heart tissue, plasma	RvD1 supplementation restored RvD1: LTB4 ratios and reduced markers of oxidative stress and necrosis.	[88]
	AT-NPD1	LC-triple quadrupole	Mouse Brain Tissue	AT-NPD1 administration 3 h post-stroke improved neurologic scores up to 7 days after stroke, reduced radiographic measures of cerebral edema, and decreased histopathologic infarct volume	[93]
	Statins	LC- triple quadrupole	Human Serum	Promotes synthesis of pro-resolving SPMs	[105]
Stroke	None	LC- triple quadrupole	Human endarterectomy plaques, mouse artery lesions	SPMs, such as RvD1 is significantly decreased in vulnerable plaque regions	[88]
	None	LC-quadrupole Orbitrap	Human Serum	FA levels vary greatly post-stroke compared to healthy controls. Phosphoglyceride profiles are distinctly different between small artery and large artery occlusions.	[89]
	None	LC-Shotgun MS	Mouse cerebral cortex	PC levels are reduced within first 7 days post-stroke, suppresses microglial secretion of pro-inflammatory cytokines. LPC levels are increased within first 7 days post-stroke, which suppresses neuronal viability.	[90]
	None	LC-Orbitrap	Human serum, Rat and Mouse cerebral cortex	plasma ceramide and sphingomyelin are increased 24 h post-stroke	[91]
Healthy	Low-dose Aspirin	LC-triple quadrupole	Human Serum	Global decrease in linoleic acid and oxylipid metabolites produced by cytochrome P450.	[94]
Alzheimer’s	DHA	LC-triple quadrupole	Human Neural Cell Line and Human Brain Tissues	DHA and NPD1 were reduced in Alzheimer’s. DHA stimulated NPD1 biosynthesis and attenuates amyloid-β secretion in cells.	[125]
	DHA	GC-MS	Human Cerebrospinal fluid	DHA increased 28%, EPA increased 43%, and EPA was 3-fold higher in non-APOE4 patients.	[149]
	None	LC-triple quadrupole	Mouse Brain Tissue	Increased AA and metabolites indicating activation of group IV isoform of phospholipase A2.	[151]
	None	LC-triple quadrupole	Human Brain Tissue	Increased 4HNE-GSH conjugates in patient temporal cortex, frontal cortex, and substantia innominata.	[129]
Parkinson’s	Levodopa	GC-MS	Human Plasma and Urine	Plasma F2-isoprostanes, HETEs, hydroxycholesterols, 7-ketocholesterol, and neuroprostanes were elevated in patients. Total HETEs was negatively correlated with levodopa intake.	[140]
COPD	None	LC-triple quadrupole	Human Serum	Identified potential biomarkers and achieved high sensitivity and specificity using a combination of 4 individual lipids and 10 lipid ratios. Increased C16:1 CE and TAG (54:6) 22:6/16:0/16:0. Decreased PI (36:6) and PI (44:6)	[178]
SARS-CoV-2	None	LC- triple quadrupole	Human Serum	Moderate and severe infections can be separated by changes in PUFAs. Changes corresponded with decreased ALOX12 and COX2, specifically loss of RvE3 and prostaglandins, and increased ALOX5 and cytochrome p450 activity in severe patients.	[179]
	None	LC-triple quadrupole	Human Bronchoalveolar Lavage	Found increased PGE2, TXB2, 12-HHTrE, and LTB4 which correlated with cytokines	[185]
	None	LC-triple quadrupole	Human Bronchoalveolar Lavage	Severe patients requiring intubation had elevated eicosanoids including thromboxane, prostaglandins, and leukotrienes (LTB4 and LTE4). SPMs increased including lipoxin A4 and D-series resolvins.	[174]
	Remdesivir	LC- triple quadrupole	Rat Plasma	DHA, RvD2, 5-HEPE, and 5-HETE levels decreased following remdesivir while TXB2 increased and PGE2 positively correlated with remdesivir metabolite concentrations in plasma.	[193]
Lung Injury	Bleomycin	LC-Orbitrap	Mouse Plasma and Bronchoalveolar Lavage	Lung samples but not plasma samples revealed changed lipid profiles. Prostaglandins increased by day 2 and ALOX5/15 DHA metabolites increased by day 7 post-injury.	[187]
	Radiation	MALDI-TOF/TOF, Orbitrap, FT-ICR MS	Rhesus Macaques Lung Tissue	Regardless of pathological findings, lipidomics identified decreased pulmonary surfactant lipids, particularly PC (14:0/16:0), PC (16:0/16:0), PC (16:0/16:1). Tissues with high histological inflammation showed high concentrations of PUFA containing PCs.	[167]
Allergic Airway	Dexamethasone	LC- triple quadrupole	Mouse Serum and Lung Tissue	Ovalbumin sensitization model induced upregulation of PGD2, PGE2, and DHA-derived protectins and 17-HDHA in lung samples but not serum. Dexamethasone activated the 17-HDHA pathway and increased protectins within 6 h.	[197]
SLE	None	GC-MS, LC-triple quadrupole	Human Plasma	TG increased, PE and PC decreased. Plasmenyl-PE has an antioxidant role which supports normal cellular functions and hence could be used as a potential biomarker.	[200]
	None	GC-MS, LC-TOF/MS	Human Plasma	Lower levels of oleic acid and EPA were associated with higher disease severity in SLE patients.	[203]
IBD	None	LC- triple quadrupole and LC-QTOF/MS	Human Plasma	Lowered EPA levels	[206]
	None	GC-MS	Mucosal membrane	AA, DPA and DHA increased	[207]
	None	LC equipped with diode array detector	Colonic Mucosa	Inhibited SPM production leading to worsening of colitis in mice.	[210]
	Dextran Sulfate Sodium	LC-triple quadrupole	Mouse Colon Tissue	Better outcome predicted with higher levels of NPD1, NPD1-isomer ((10S,17S)-DiHDoHE) reduced neutrophil infiltration and inflammatory markers	[214]
	None	GC-MS and LC-MS	Human Plasma	PUFA and eicosanoids derived from AA corresponded to increased colonic inflammatory cytokines found in the bowel inflammation process	[220,221,222]

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
