# Peer review of "Lipidomics in Understanding Pathophysiology and Pharmacologic Effects in Inflammatory Diseases: Considerations for Drug Development"

_metabolites, 2022, doi:10.3390/metabo12040333_

Round 1

Reviewer 1 Report

This is a comprehensive and general review about the state of the art of lipidomics in the context of diseases with an inflammatory component. Although the domain is vast and complex, the authors have constructed a quite balanced and accurate manuscript.

Comments:

  • The title is too general and does not correspond to the contents. It could be completed with something like “in inflammatory diseases” because I think this is the focus of the review. Also, the review is not only about pharmacologic effects and drug development, but also on pathophysiology.
  • When providing a list of MS approaches (page 3 and table 1), LC-MS/MS ( a very general acronym) is listed along with specific technologies. In fact, all or most of the technologies can be defined as LC-MS/MS. I wonder if what authors mean by LC-MS/MS is LC-triple quadrupole. The latter would be a better definition.
  • HPLC-ELSD is listed as an MS technology. However, HPLC-ELSD does not involve mass spectrometry.
  • On page 7, three long paragraphs (lines 254-292) do not contain any references, in spite of the amount of information provided. Some references should be added.
  • Line 347: “and other cholesterol esters”. I guess the authors mean “and other sources of cholesterol esters” or something similar.
  • Line 694: “have identify” should be replaced by “have identified”.
  • Please include a list of abbreviations.
  • A suggestion: a table at the end of the manuscript recapitulating the main changes in metabolites in the different diseases addressed in the text.

Reviewer 2 Report

Review for the manuscript Lipidomics in Understanding Pharmacologic Effects and Drug 2 Development

Dear Editor, thank you very much for the opportunity to review this interesting and impressive manuscript.

The authors aimed to build considerations for “lipidomics study designs including instrumentation, sample stabilization, data validation, and data analysis”. In addition, the authors intended to “review highlights how lipidomics can be applied to biomarker discovery and drug mechanism dissection in various inflammatory diseases including cardiovascular disease, neurodegeneration, lung disease, and autoimmune diseases”. Although the topic is relevant, I suggest that the authors perform minor revisions before the manuscript can be published.

INTRODUCTION

In lines 40-43, the authors say that “Additionally, lipidomic profiling has identified imbalances in lipid homeostasis in several diseases including metabolic syndrome, cardiovascular disease (CVD), neurodegenerative disease, respiratory disease, and auto-immune disease [1,2]”. I think it would be interesting for the authors to point out some examples of how lipidomics could help in these conditions that are of global interest. For example, metabolic diseases are among the most deadly diseases in the world. Immune diseases have been growing sharply around the globe. Therefore, I suggest exploring more information in this paragraph.

Furthermore, I suggest that the authors improve/extend the definitions of metabolomics and lipidomics as this subject is still new to many readers. I can see that there is information on these issues throughout the manuscript, but I believe the reader would be more comfortable finding deeper definitions already in the introduction.

I also suggest including some newer references in this section.

Lines 96-97: the authors say that” At the center of these metabolomics/lipidomics analyses is the use of mass spectrometry (MS) technology”. Are there other technologies to evaluate metabolomics/lipidomics analyses? If yes, explain the reason to use MS. Please, briefly discuss the costs of this technique.

In lines 178-180 we can read:

“Lipidomics can identify analytes against a large and complex background, where both experimental or environmental factors, such as age, diet, growth phase, media, nutrients, pH, sex, and temperature can play roles in metabolite concentrations”.

I suggest modifying for “Lipidomics can identify analytes against a large and complex background, where both experimental or environmental factors, such as age, diet, growth phase, media, nutrients, pH, sex, ethnicity, genetic factors, and temperature can play roles in metabolite concentrations”.

In Lines 226-228 we can read: “lipidomic analyses can be aimed at certain subgroups of lipid species such as phospholipids, polyunsaturated fatty acids (PUFA) and their bioactive metabolites (eicosanoids and docosanoids), steroids, and triglycerides.”

I suggest modifying for “lipidomic analyses can be aimed at certain subgroups of lipid species such as steroids, triglycerides, phospholipids, polyunsaturated fatty acids (PUFA) and their bioactive metabolites (eicosanoids and docosanoids)”.

Please, include IL-6 and IL-10 in the legend of Figure 1.

In Lines 414-417, please change “Depuydt et al were able to identify distinct phenotypic subclasses within each immune cell population, highlighting the cellular plasticity and complex intercellular interactions at the disease site [71].” For “Depuydt et al [71] was able to identify distinct phenotypic subclasses within each immune cell population, highlighting the cellular plasticity and complex intercellular interactions at the disease site.”

I also suggest changing “As recently reviewed by Miao et al., several other SPMs, including LXs and maresins (MRs), have been studied as potential therapeutics after stroke [88].” For As recently reviewed by Miao et al. [88], several other SPMs, including LXs and maresins (MRs), have been studied as potential therapeutics after stroke” (Lines 474-476].

Please, perform this kind of correction along with the entire text.

Do authors think it would be interesting to include in line 670 a sub-section named Parkinson´s Disease?

Before the Conclusions, I suggest that the authors include a section showing the limitations of this review.
